# Integral methods for automatic quantification of fast-scan-cyclic-voltammetry detected neurotransmitters

**Leonardo X. Espín**[1☯], **Anders J. Asp**[2☯], **James K. Trevathan**[2], **Kip A. Ludwig**[1], **J. Luis Lujan**[1,3]*

**1** Department of Neurologic Surgery, Rochester, Minnesota, United States of America, **2** Mayo Clinic Graduate School of Biomedical Sciences, Rochester, Minnesota, United States of America, **3** Department of Physiology and Biomedical Engineering Mayo Clinic, Rochester, Minnesota, United States of America

☯ These authors contributed equally to this work.
* lujan.luis@mayo.edu

**Data Availability Statement:** Data and code are available on GitHub (https://github.com/aspanders/FSCVIntegralMethods).

## Abstract

Modern techniques for estimating basal levels of electroactive neurotransmitters rely on the measurement of oxidative charges. This requires time integration of oxidation currents at certain intervals. Unfortunately, the selection of integration intervals relies on ad-hoc visual identification of peaks on the oxidation currents, which introduces sources of error and precludes the development of automated procedures necessary for analysis and quantification of neurotransmitter levels in large data sets. In an effort to improve charge quantification techniques, here we present novel methods for automatic selection of integration boundaries. Our results show that these methods allow quantification of oxidation reactions both *in vitro* and *in vivo* and of multiple analytes *in vitro*.

## 1 Introduction

Fast scan cyclic voltammetry (FSCV) is a powerful electrochemical sensing technique that allows quantification of variations in the concentration of electroactive neurochemicals by measuring redox currents resulting from the application of a periodic triangular waveform at a high scan rate [1–5]. Traditionally, FSCV has depended on the calculation of maximal oxidation currents, measured from known neurochemical concentrations in a solution, which are used to build calibration curves by using linear correlation techniques [6–10]. Recent studies have exploited the catecholamine adsorption properties of carbon fiber microelectrodes (CFM), to estimate basal concentrations of neurochemicals [11–14]. Techniques including fast scan cyclic adsorption voltammetry (FSCAV) use oxidation-charge measurements, rather than maximal currents, which are obtained by time-integrating cyclic voltammograms within intervals containing single oxidation peaks (or "humps") [11–13]. However, the accuracy and reproducibility of oxidation-charge measurements are limited by visual selection of integration bounds of the cyclic voltammogram oxidation peaks. In practice, defining which portion of the voltammogram constitutes an oxidation peak (where it begins, and where it ends) is

**Funding:** This work was funded by the National Institutes of Health grant R01 NS084975 awarded to J. Luis Lujan. The funders had no role in study design, data collection and analysis, decision to publish, or preparation of the manuscript. https://www.nih.gov/.

**Competing interests:** The authors have declared that no competing interests exist.

obscured by the noise floor of the dataset, the electrochemical interferents, the presence of artifacts, and background drift. Visual selection leads to ambiguity, introduces additional sources of error and precludes the development of automated procedures necessary for analysis and charge quantification in large data sets.

Here, we describe novel charge quantification techniques by performing automatic selection of integration boundaries. This is achieved by analyzing and identifying voltammogram's critical, inflection and maximum curvature points, to allow the automatic selection of integration intervals. We test these techniques in both *in vitro* and *in vivo* experimental scenarios.

## 2 Methods

### 2.1 In vitro data collection

The Mayo Clinic Institutional Animal Care and Use Committee (IACUC) has reviewed and approved this research. Anesthesia and euthanasia were performed per Mayo Clinic IACUC regulations with urethane and fatal plus, respectively. In vitro data collection was performed using a FIAlab 3200 flow injection system (FIAlab Instruments, Seattle, WA) and the WINCS Harmoni device [15]. A CFM of 7μm diameter and110μm in length was placed in a flowing stream of artificial cerebrospinal fluid (aCSF) buffer solution with a pH value of 7.4 as described previously [15]. For each measurement, buffered aCSF solutions containing 0.1 $\mu$M to 5 $\mu$M of either dopamine, adenosine, epinephrine, or norepinephrine (Millipore Sigma, Burlington, MA) were injected for 8 s at 2.25 mL/min. The *in vitro* data was collected using two electrodes, with 3–15 injections per analyte.

### 2.2 In vivo data measurements

In vivo measurements were obtained in a rodent model of medial forebrain bundle stimulation and simul- taneous FSCV recording in the dorsal striatum as described previously [15]. Rats were sedated prior to surgery with intraperitoneal urethane (1.5 g/kg in a 0.26 g/mL saline solution, Millipore Sigma, Burling- ton, MA). Analgesia was maintained for rodents with intramuscular buprenorphine (0.06 mg/kg). Both the stimulation electrode and CFM were ste- reotactically inserted (KOPF instruments, Tujunga, CA). A scalp incision (1.5–2.0 cm) was made to expose the skull, and three trephine burr holes (approximately 3 mm in diameter) were drilled to allow implantation of the stimulating, neurochemical sensing, and reference electrodes. Dopamine release was evoked with a series 2 s stimulations using a range of amplitudes from 0.05–2.0 mA and pulsewidths from 0.8 and 1.8 ms, presented in a randomized order such that effects of hysteresis are minimized. Analyte measurements were obtained by sweeping the CFM potential from a resting potential of -0.4 V to a switching potential of 1.5 V and back to the resting potential, at a rate of 400 V/s every 100ms.

### 2.3 Charge quantification

We define the charge resulting from a single oxidation/reduction reaction by

$$Q = \int_{x_l}^{x_r} B(t)dt, \qquad (1)$$

where $B(t)$ is a background subtracted voltammogram, and we assume that the integration interval $[x_l \, x_r]$ contains a single oxidation/reduction peak (Fig 1a). In this study we use four pairs of integration boundaries (Fig 1) to quantify charge. When there is little or no background drift, indicating a stable background capacitive current, the faradaic current response on a background subtracted voltammogram decays towards zero away of the maximum oxidation current [16]. "True" integration boundaries are defined as the points around an oxidation

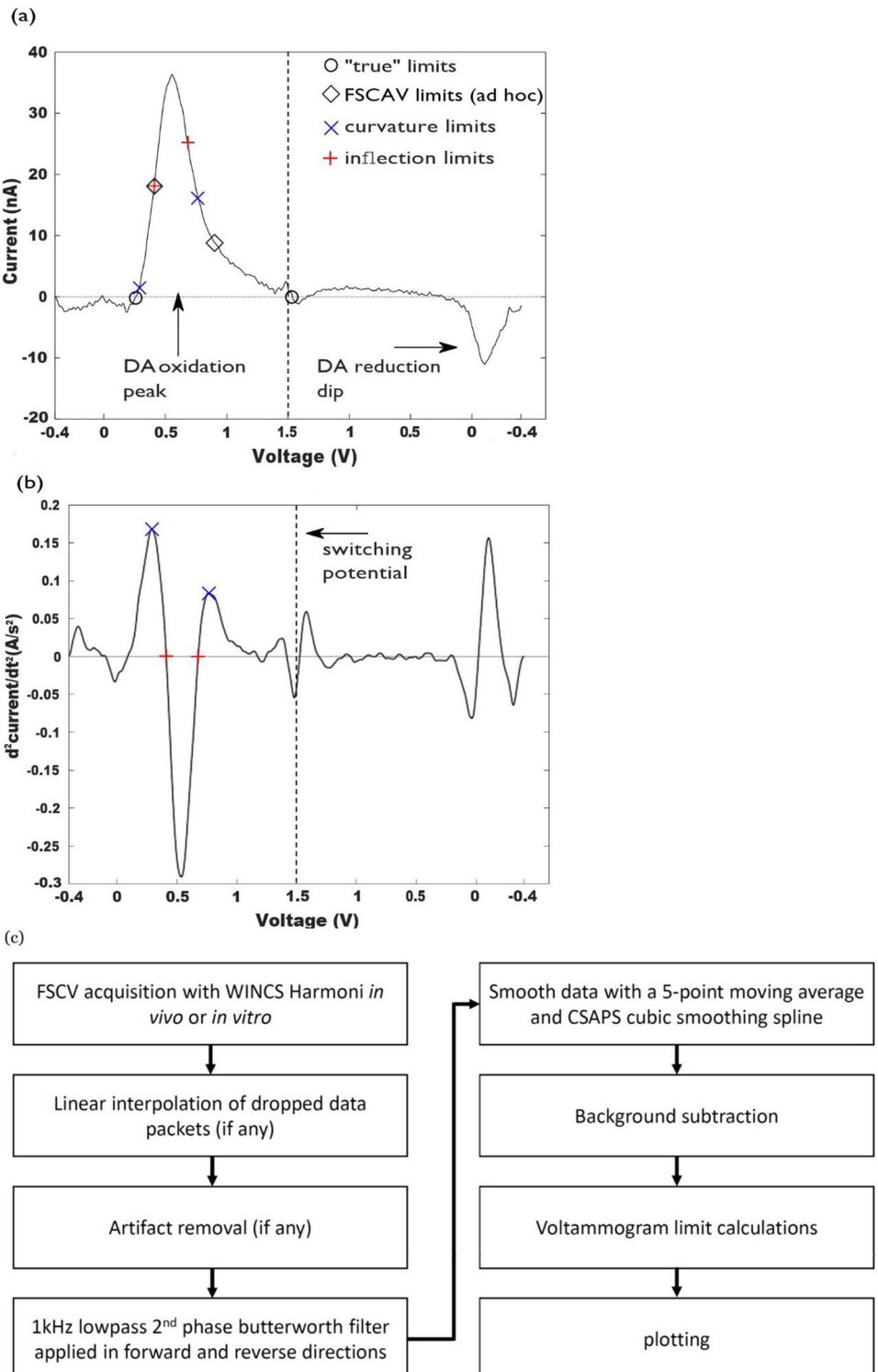

**Fig 1.** a) Background subtracted voltammogram for a bolus of a 0.5 $\mu$M dopamine solution. True ("o", $x_l = 0.26$ V and $x_r = 1.5$ V), FSCAV ("♦", $x_l = 0.4$ V and $x_r = 0.9$ V), inflection ("+", $x_l = 0.41$ V and $x_r = 0.68$ V) and curvature integration boundaries ("x", $x_l = 0.29$ V and $x_r = 0.76$ V) are marked on the voltammogram. b) Second derivative of the voltammogram shown in (a), showing the definition of inflection boundaries (inflection points), and curvature boundaries (the maxima around the location of the oxidation peak). c) Flow chart demonstrating overview of applied computational procedures to raw FSCV data.

peak where the current is zero (Fig 1a). Charges computed with true boundaries provide a useful benchmark for comparing quantification methods. However, "True" limits as defined here are unavoidably confounded by the interplay between the Gaussian dopamine oxidation peak and the noise levels in the recording; as a Gaussian distribution never decays to zero.

FSCAV limits (Fig 1a), correspond to the ad-hoc voltages (0.4 V and 0.9 V) selected as integration limits for the quantification of dopamine [13]. Note that all measurements were collected using a traditional FSCV waveform described in section 2.2 as opposed to FSCAV. Inflection limits are defined as the zeros of the second derivative of a voltammogram, around an oxidation (or reduction) peak, and curvature points (maximum curvature, or maximum second derivative) are defined as the maxima/minima of the second derivative around an oxidation/reduction peak, see Fig 1(b).

Computational routines for filtering and smoothing background- subtracted voltammograms, as well as to calculate higher order derivatives, to find, classify, sort and correct curvature and inflection points are implemented in MATLAB (Fig 1). A 1000Hz lowpass $2^{nd}$ phase butterworth filter was applied to all data in both forward and reverse directions to reduce spurious signals. Data was also smoothed with a 5-point moving average and a CSAPS Cubic smoothing spline before background subtraction subsequent charge calculations. Correction routines require not adding negative areas to the total charge in the case of oxidation reactions, and have the switching potential as hard limit for $x_r$. Similar considerations are utilized for reduction-charge calculations (excluding regions with positive areas, and using the switching potential as hard limit for $x_l$).

## 3 Results and discussion

### 3.1 Quantification of in vitro catecholamines

In Fig 2 we show (a) true and (b-c) curvature integration boundaries obtained with our algorithms. As a reference we also show the location of the maximum oxidation currents with dots, and panels (b- c) show the FSCAV integration limits taken from reference [11], 0.4 V and 0.9 V with vertical dashed lines. The data for panels (a) and (b) corresponds to background subtracted voltammograms of a flow cell experiment, where 70 dopamine injections with 0.1, 0.5 and 1 micromolar were done. The data for panel (c) corresponds to voltammograms of a flow cell experiment, where 15 norepinephrine injections with 0.1, 0.5, 1 and 5 micromolar were done.

Panels (a) and (b) of Fig 2 show that the curvature boundaries select narrower integration intervals than true boundaries. However, it is interesting to note that curvature boundaries show less variability than the true boundaries (average voltages and standard deviations are shown in Fig 2). This is unex- pected given that the computation of curvature boundaries involves calculating higher-order derivatives of voltammograms, which should amplify noise [17], and as a consequence curvature boundaries variability.

Because of the visual similarities between typical dopamine and norepinephrine background subtracted voltammograms [4,11,18], comparisons using norepinephrine data are particularly useful, because they highlight the limitations of using visual identification to obtain integration boundaries. Indeed, panels (b) and (c) of Fig 2 indicate that there is a significant difference between average left curvature boundaries for dopamine and norepinephrine voltammograms.

Indeed, Fig 3 shows oxidation-charge calculations with the four pairs of integration boundaries intro-duced in section 2.3. Panel (a) shows calculations for the dopamine dataset corresponding to (Fig 2a and 2b), and panel (b) shows calculations for the norepinephrine dataset corresponding to Fig 2(c). Fig 3(a) shows that when quantifying dopamine charge, the ad-hoc

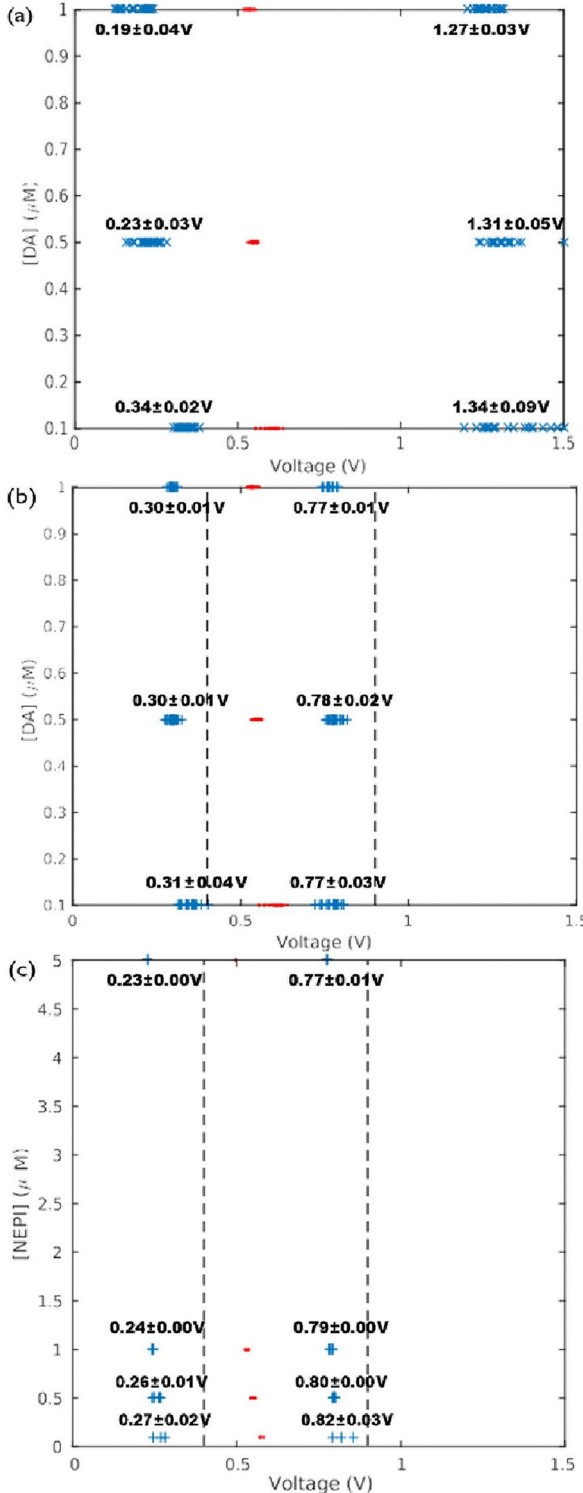

**Fig 2.** a) True integration limits ("x") and maximum oxidation currents (dots) obtained from a flow cell experiment consisting of 70 dopamine injections at different concentrations. b) Curvature integration boundaries ("+") for the experimental data of (a). c) Curvature boundaries for a flow cell experiment with 15 norepinephrine injections at different concentrations. FSCAV limits are shown with dashed vertical lines. Quantities shown in the figure indicate averages ± standard deviations.

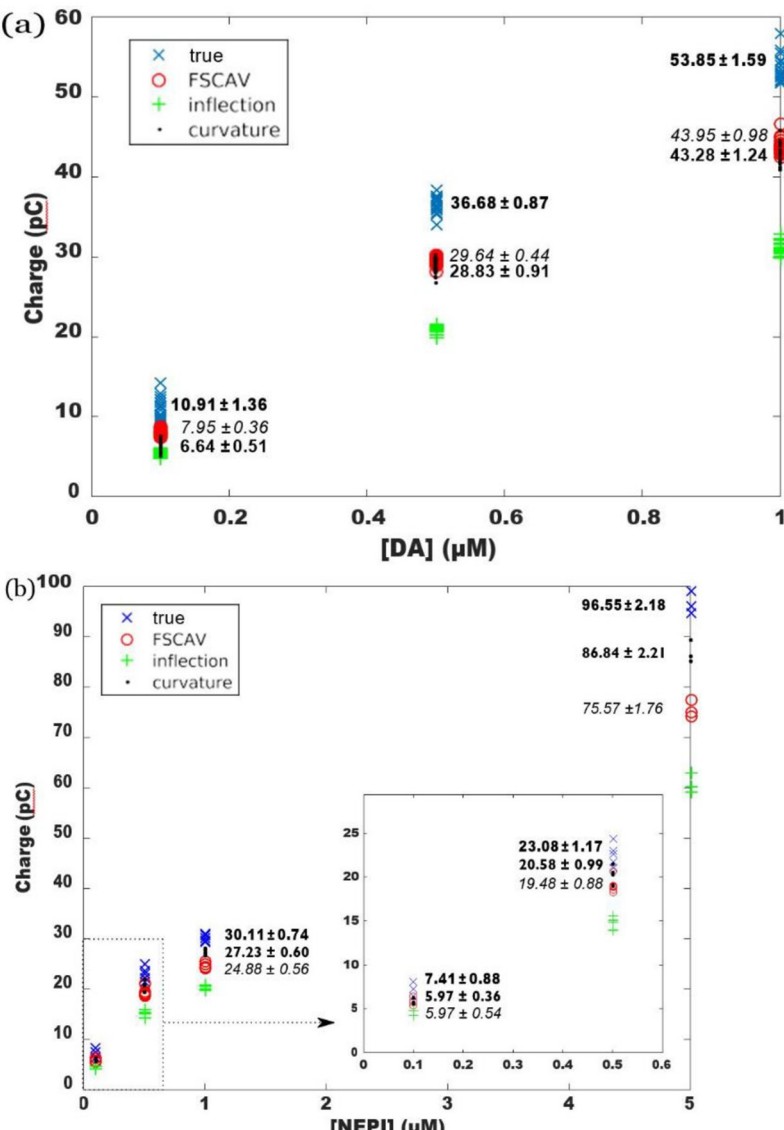

**Fig 3.** a) Oxidation charges for the data shown in (Fig 2a and 2b), obtained with different integration boundaries. b) Oxidation charges for the data shown in Fig 2(c), obtained with different integration boundaries. Averages and standard deviations for charges using true, FSCAV (in italic) and curvature (bold) integration boundaries are also shown. Quantities shown in the figure indicate averages ± standard deviations.

FSCAV limits and curvature are not significantly different at concentrations greater than 0.1μM as demonstrated by a comparison of mean curvature and FSCAV datapoints using an ANOVA with Holm-Sidak comparison for multiple corrections, P>0.05 n.s. However, despite shape similarities between dopamine and norepinephrine background subtracted voltammograms (e. g. correlation coefficient > 0.86, see [18,19]), oxidation charges obtained with curvature limits provide a closer approximation to true charges than FSCAV limits at concentrations of 1μM and greater, as demonstrated by a comparison of mean curvature and FSCAV datapoints using an ANOVA with Holm-Sidak comparison for multiple corrections (at 0.5μM, curvature (20.58±0.99) v. FSCAV(19.48±0.88), P = 0.1087 n.s.; at 1μM, curvature (86.84±2.21) v. FSCAV(75.57±1.76), P<0.0001 ****; at 5μM, curvature (27.23±0.6) v. FSCAV

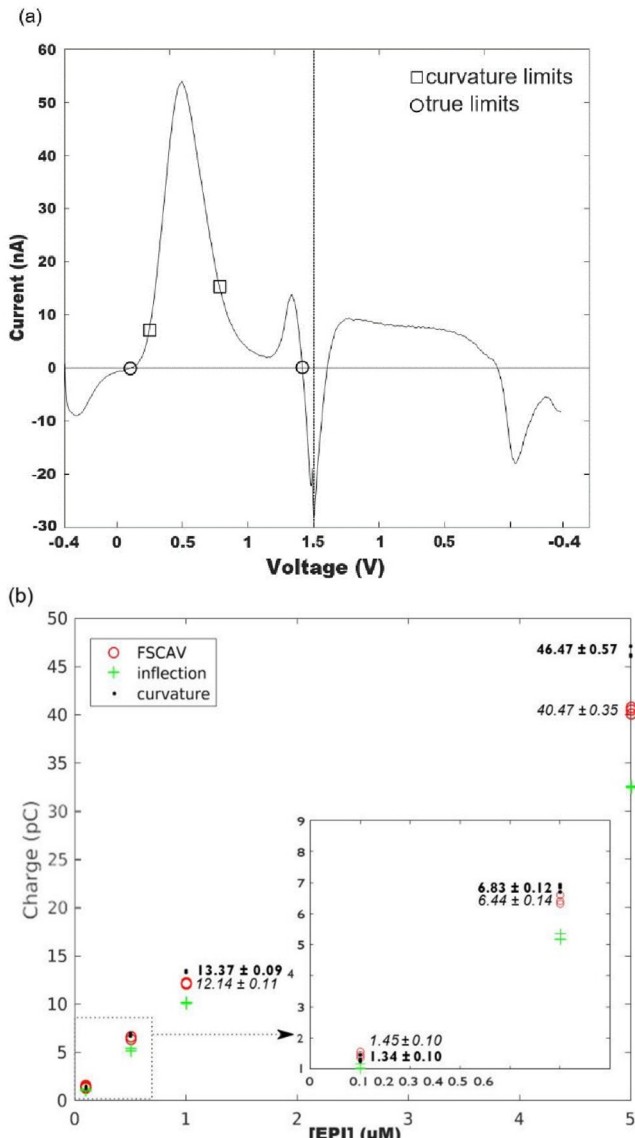

**Fig 4.** a) A 1 $\mu$M epinephrine voltammogram showing primary and secondary oxidation peaks, superimposed with curvature and true integration limits. b) Primary-peak oxidation charges for a flow cell experiment with 15 injections of epinephrine at different concentrations, obtained with different pairs of integration limits. Averages and standard deviations for charges using FSCAV (in italic) and curvature (bold) integration boundaries are also shown. Quantities shown in the figure indicate averages ± standard deviations.

(24.88±0.0.56), P<0.01 ****) (Fig 3b).” Consequently, all analyses of charge in subsequent figures are performed using the curvature method in the interest of focus and clarity.

Curvature integration boundaries provide the highest charge among the three methods considered at concentrations above 0.5 μM as shown by a comparison of curvature and FSCAV datapoints using an unpaired two-tailed t-test with welch's correction (0.1μM, p>0.05; 0.5μM, p = .024; 1μM, p = .0001; 5 μM, p = .0003) (Fig 4).”As illustrated by Fig 4(a), epinephrine voltammograms have two oxidation peaks [3], and if we try to calculate the oxidation charge due to the primary (or secondary) peak alone, true boundaries produce erroneous

results by selecting a region that contains both oxidation peaks. This issue also arises when combinations of analytes (like dopamine and adenosine) are being measured.

## 3.2 Charge quantification of in vivo measurements

Rapid changes in the brain electrochemistry can lead to faster background current drift [2,20,21], which distorts the voltammograms (an example is shown in Fig 5a). The use of true charges as a benchmark for charge quantification depends on the stability of the background current measured with FSCV. Thus, faster in-vivo background-current change can pose a problem for the use of true boundaries with in-vivo data, as we explain below. Fig 5(a) shows a stimulation-evoked background-subtracted voltammogram from a rodent striatum dopamine measurement evoked by systematically varying amplitude and duty cycle applied in a random order to minimize hysteresis., superimposed with curvature boundaries. A region of positive, nearly constant current between 1 and 1.5 volts that persists far into the *cathodic* sweep, is likely the consequence of background drift. Drift is exacerbated by disruption of Helmholtz layer or pH changes as a result of electrical stimulation [22]. The integration boundaries can still be calculated in the presence of background drift, though the exact cutoffs may be affected. It is generally advisable for the experimenter to minimize any possible sources of background drift or other redox sources in FSCV recordings for consistent data analysis and interpretation. The shaded region adds a positive bias to the charge computed with true boundaries. Furthermore, a similar behavior is observed in most voltammograms of the data set of Fig 5. Consequently, in panel (b), which shows true and curvature boundaries for the entire dataset, the right true boundary $x_r$ has been set to the switching potential of the voltage sweep.

We contrast the behavior of true boundaries for the experimental data set of Fig 5(b) with that of the curvature boundaries, which despite the random variation of the DBS stimulation parameters of the experiment, show little variability ($x_l = 0.388 \pm 0.007$ V and $x_r = 0.807 \pm 0.022$ V), demonstrating the robustness of our charge quantification algorithms.

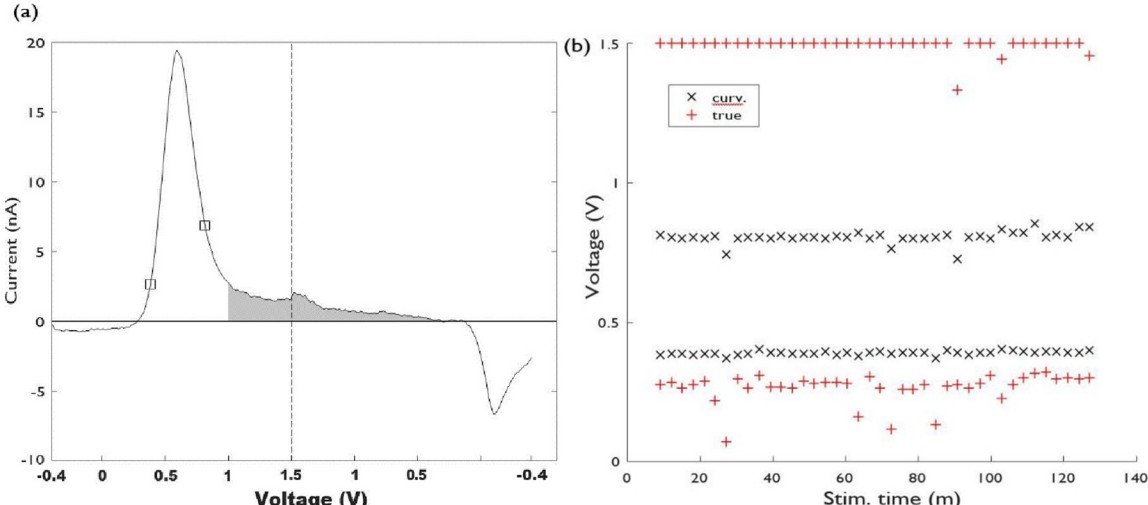

**Fig 5.** a) Background subtracted voltammogram of a dopamine signal taken from a rat during DBS of the medial forebrain bundle (0.14mA, 1.43ms pulse width, 180 pulses, 90Hz), superimposed with squares showing the curvature boundaries for the voltammogram. Charge in the shaded region is likely the result of background current drift. Stimulation amplitude and pulse widths ranging from 0.14–0.16ms and 1.37–1.57ms, respectively, were systematically applied in a random order to minimize effects of hysteresis. All stimulation epochs consisted of 180 pulses delivered at 90Hz b) True and curvature boundaries obtained from background subtracted voltammograms of the entire rat DBS dataset with random combinations stimulation.

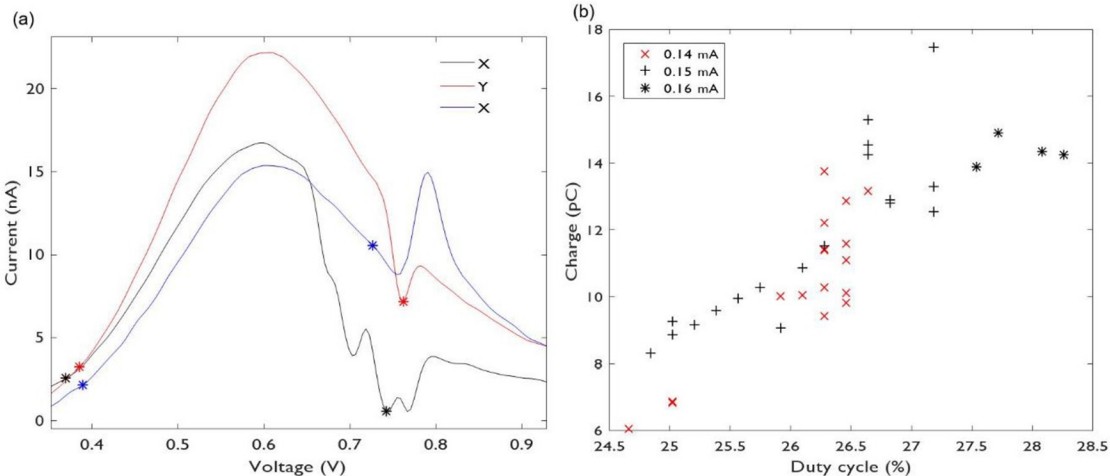

**Fig 6.** a) Voltammogram regions enclosing the oxidation peak of selected background subtracted voltammograms taken from the data set of Fig 5, superimposed with asterisks showing the curvature boundaries for each voltammogram. Stimulation parameters for the traces X,Y, and Z are as follows X: 0.14mA, 1.46ms pulse width, 180 pulses, 90Hz Y: 0.16mA, 1.57ms pulse width, 180 pulses, 90Hz Z: 0.15mA, 1.45ms pulse width, 180 pulses, 90Hz (b) Oxidation charges corresponding to the data set of Fig 5, obtained with curvature limits, plotted as function of the duty cycle for the 90 Hz, two-second stimulations.

This robust behavior can be utilized for outlier detection purposes. Values that deviate the most from the average right curvature boundary of the voltammograms in ($x_r$ = 0.807 ± 0.022 V) Fig 5(a) indicate that stimulation artifacts have altered the shape of the voltammograms and the corresponding curvature boundaries have adapted to the shape of each curve 6(a). Here, details of the oxidation-hump region of three voltammograms are shown with their corresponding curvature boundaries indicated by asterisks, demonstrating that the integration limits can be objectively determined with varying stimulation parameters, even in the presence of sources of noise such as a stimulation artifact.

In Fig 6(b) we present the oxidation charges corresponding to the in-vivo experimental data set of Fig 5, as function of the stimulation duty cycle. This panel shows how the response measured by the CFM increases as a function of the increasing stimulation duty cycle.

### 3.3 Quantifying charge produced in reduction reactions

As described in section 2.3, our methods can be used to quantify charged due to reduction reactions, and our algorithms require minimal alterations to do so (finding minima instead of maxima, etc).

Fig 7(a) shows the curvature limits for the reduction dips measured in the data set of Fig 3(a). Fig 7(b) we show the corresponding charges obtained with curvature limits, as well as with inflection limits. We note that for reduction reactions, very often a right true limit is not existent (see Fig 8(a) for an example). In consequence charges obtained with true limits are not shown.

### 3.4 Quantifying charge in voltammograms with multiple oxidation peaks

If multiple electroactive analytes are present in a FSCV measurement, quantifying oxidation due to different species can be challenging. In this section we illustrate how our quantification techniques can aid in that task, when used with cyclic voltammograms displaying multiple humps. Linear charge calibration data for dopamine, adenosine, and a solution of

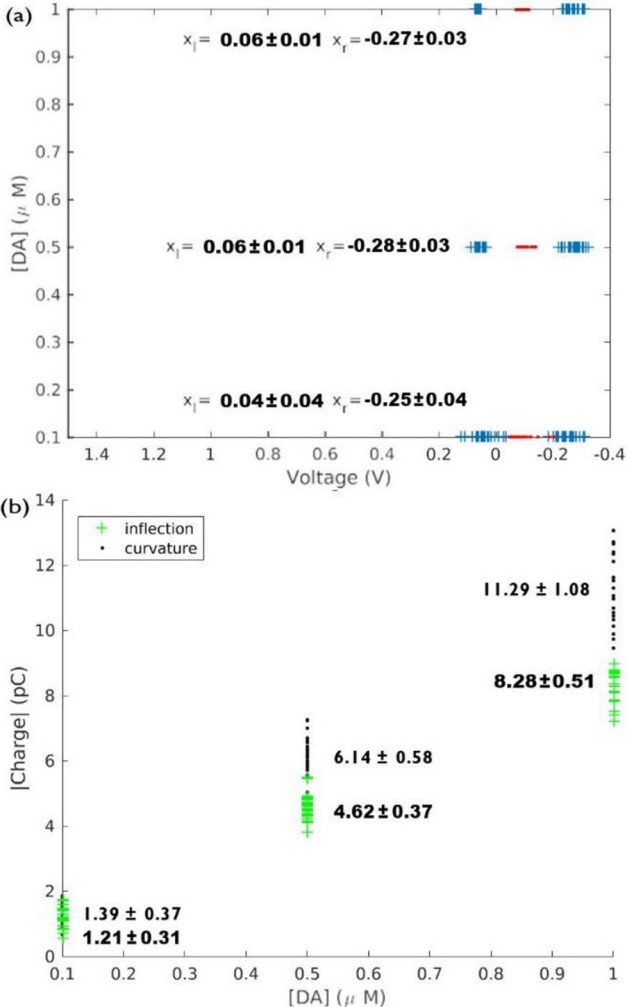

**Fig 7.** a) Curvature integration limits for the reduction dips of the data set of Fig 3(a). Minimum reduction currents are shown with asterisks. b) Reduction charges (absolute value) for the data set of Fig 3(a), obtained with inflection and curvature (bold) integration boundaries. Quantities shown in the figure indicate averages ± standard deviations.

dopamine+adenosine were calculated using true, inflection, and curvature integration boundaries (S1 Fig). Indeed, the definitions of curvature limits and inflection limits can be used for individual oxidation humps, as we show in Fig 8, where two examples of multiple-oxidation-hump voltammograms, as well as curvature boundaries for each peak are displayed. Notice that in Fig 8(a) the two peaks appear because of two different species, while the voltammogram of Fig 8(b) is the result of the oxidation of epinephrine (see also Fig 4(a)).

## 3.5 Challenges and limitations

While this approach may be helpful for standardizing oxidation charge quantification of electroactive neurochemicals, it is not without limitations. The relationship between charge and analyte concentration is nonlinear (Fig 3). While linear regressions are regularly used when creating a calibration curves [6–10], this approach may result in underestimations at the low and high ends of detectable concentrations while overestimations concentrations at the

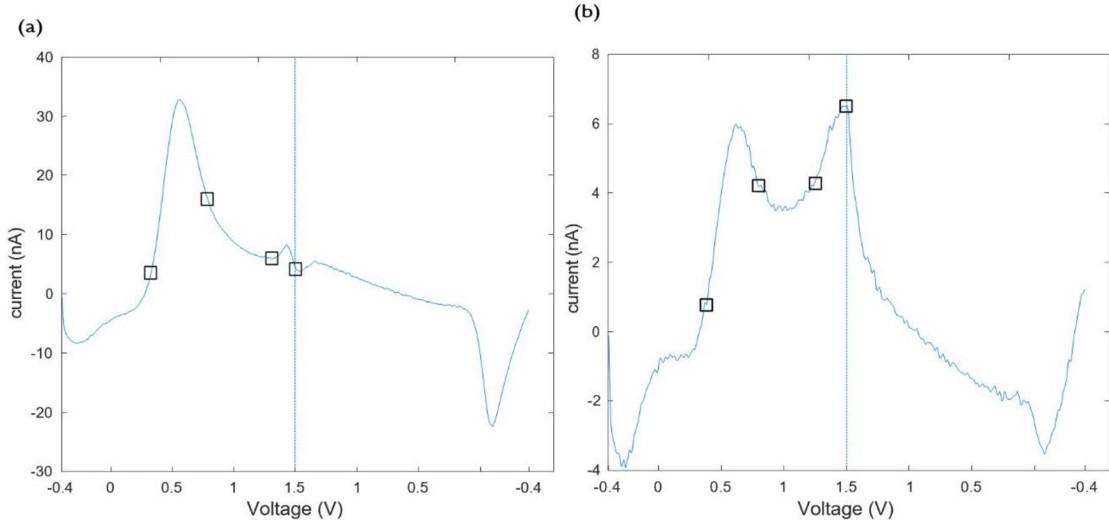

**Fig 8.** a) Background subtracted voltammogram showing two oxidation peaks due to 1 $\mu$M dopamine plus 1 $\mu$M adenosine, along with the corresponding curvature boundaries for each peak: 0.33 V and 0.79 V for DA, and 1.31 V and 1.5 V for ADO. b) Background subtracted voltammogram showing two oxidation peaks due to 0.5 $\mu$M epinephrine, and the corresponding curvature boundaries for each peak: 0.38 V and 0.80 V for first peak, and 1.25 V and 1.5 V for the second. Vertical dashed lines indicate the switching potentials.

midpoint of the detectible range. Fitting with a nonlinear regression model may reduce these errors when converting between charge and concentration for unknown samples. Further-more, we present a limited analysis of the performance of automated charge calculation meth-ods in a sample of multiple analytes, but a more exhaustive characterization of charge in the presence of other electroactive interferents (e.g. DOPAC, Guanine, Serotonin, etc.) is neces-sary to fully determine performance of this approach. Despite these limitations, computational approaches such as the one described here are necessary for improving accuracy and consis-tency of electroactive neurochemical measurements.

## 4 Conclusions

Here, we propose novel methods to quantify charge from REDOX reactions observed in vol-tammetric measurements electroactive neurochemicals. While our proposed methods were applied to a dataset collected with a traditional FSCV waveform, this approach can be gener-alized to other voltammetric waveforms reliant on charge quantification, including FSCAV. However, additional studies must be completed to characterize behavior of our approach with novel voltammetric waveforms. Although a definitive selection of integration bound-aries is confounded by interferents, background drift, low signal to noise ratio, and other effects, data shown here suggest that our quantification methods are comparable to compet-ing methods for quantification of dopamine oxidation charge, and may more closely approx-imate true charge when applied to other catecholamines. Additionally, unlike existing charge quantification techniques, our methods can quantify reduction reactions as well as single oxidation or reduction peaks when multiple analytes are present in a sample. Here, charge analysis has been performed automatically, improving reproducibility and demon-strating the feasibility of developing automated routines for charge quantification of multi-ple analytes.

## Supporting information

**S1 Fig. Linear charge calibration curve for dopamine, adenosine, and dopamine +adenosine using a)True integration limits b) curvature integration limits, and c) inflection integration limits at concentrations of 0.1 μM, 0.5 μM, and 1 μM.** Data are fit with a linear regression and $R^2$ values are displayed for each analyte.
(DOCX)

## Author Contributions

**Conceptualization:** Leonardo X. Espín, Anders J. Asp, Kip A. Ludwig, J. Luis Lujan.

**Data curation:** Anders J. Asp, James K. Trevathan.

**Formal analysis:** Leonardo X. Espín, Anders J. Asp.

**Funding acquisition:** J. Luis Lujan.

**Investigation:** Anders J. Asp, James K. Trevathan.

**Methodology:** Anders J. Asp, James K. Trevathan, Kip A. Ludwig, J. Luis Lujan.

**Project administration:** Leonardo X. Espín, Kip A. Ludwig, J. Luis Lujan.

**Resources:** Kip A. Ludwig, J. Luis Lujan.

**Supervision:** Kip A. Ludwig.

**Validation:** Anders J. Asp.

**Visualization:** Anders J. Asp.

**Writing – original draft:** Leonardo X. Espín.

**Writing – review & editing:** Leonardo X. Espín, Anders J. Asp, James K. Trevathan, Kip A. Ludwig, J. Luis Lujan.

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
