## [Decision Letter · Decision Letter 0]

27 Oct 2020

PONE-D-20-25050

Integral methods for automatic quantification of fast-scan-cyclic-voltammetry detected neurotransmitters

PLOS ONE

Dear Dr. Asp,

Thank you for submitting your manuscript to PLOS ONE. After careful consideration, we feel that it has merit but does not fully meet PLOS ONE’s publication criteria as it currently stands. Therefore, we invite you to submit a revised version of the manuscript that addresses the points raised during the review process.

Both reviewers agree on the importance of your work, but also raised many important questions that need to be clarified. Please consider each point carefully in your revision and make it clear how you addressed each of these points in your revised version.

We look forward to receiving your revised manuscript.

Kind regards,

Maria Asplund, PhD

Academic Editor

PLOS ONE

Journal Requirements:

Reviewers' comments:

Reviewer's Responses to Questions

**Comments to the Author**

1. Is the manuscript technically sound, and do the data support the conclusions?

Reviewer #1: Partly

Reviewer #2: Yes

2. Has the statistical analysis been performed appropriately and rigorously? 

Reviewer #1: No

Reviewer #2: Yes

3. Have the authors made all data underlying the findings in their manuscript fully available?

Reviewer #1: Yes

Reviewer #2: Yes

4. Is the manuscript presented in an intelligible fashion and written in standard English?

Reviewer #1: Yes

Reviewer #2: Yes

5. Review Comments to the Author

Reviewer #1: This study discusses the development of a novel methodology for selecting integration limits for the purpose of quantifying the concentration of electrochemical targets using FSCV. This technology advancement could serve as a critical improvement to the in vivo basal detection method FSCAV by replacing user-defined integration limits with algorithm-defined cutoffs. This would have the desired effect of removing the possibility for user bias from FSCAV data analysis. This study is highly relevant but requires substantial revision prior to publication.

Major

• How does the curvature method perform when dealing with small in vivo oxidation peaks resulting from FSCAV basal detection or brief DBS stimulation? The authors state that the desired use for is for this methodology is to provide cutoff limits for in vivo FSCAV experiments. This study should show the applicability of the methodology using FSCAV data.

• The responses in Figure 3 display varying degrees of linearity (none of them highly linear). Please discuss the consequences of this given that linear regression of these responses is used to convert between charge and concentration for unknown samples. Does the degree of linearity for the various methods adjust the determination of which method is best suited for use?

• Please provide detailed statistical analysis between groups to support claims. For example, the author’s claim that there is a “significant difference between average left curvature boundaries for DA and NEP” is not supported by statistical evidence.

• There are many cases where the authors neglect discussing data sets within the figures. For example, the authors largely neglect discussing the data arising from the inflection method throughout the entire manuscript as well as the FSCAV limits for EPI. Please discuss all relevant data provided in the figures.

• The authors state that “A region of positive, nearly constant current between 1 and 1.5 volts that persists far into the cathodic sweep is likely the consequence of background drift.” Is there evidence to support that claim? If the feature is drift, why does that drift not occur prior to the oxidation peak (-0.4 V to 0.3 V)?

• What is meant by the “random variation of the DBS stimulation parameters” in the main text when describing the data in Figure 5. Were these data collected using the various duty cycles detailed in Figure 6? What set of conditions were used to produce the response in Figure 5a?

• Are the results detailed in Figure 6b meant to state that the increase in the charge is due to an increase in the extracellular concentration of the in vivo chemical (I assume the chemical is dopamine but the authors never state the identity of the in vivo signal) or is this a consequence of the duty cycle artifact on integration limits? In addition, the authors neglect discussing how the data seem to group by stimulation current as well. Please substantially revise and clarify this discussion.

• It is unclear why the authors would want to optimize quantifying stimulation artifacts in Figure 6. Stimulation timing is routinely staggered with FSCV recordings to ensure that artifacts do not distort background subtracted voltammograms. Please explain.

• The 2nd EPI peak in Figure 8b has clearly not resolved by the switching potential. Cutting at the switching potential does not seem appropriate for obtaining an accurate quantification of the 2nd peak. Is this technique only applicable for targets where the peaks are allowed to totally resolve prior to the switching potential?

• How does the sensitivity for target (DA) detection in the presence of interferent compare to the sensitivity for target alone using the various methods? This is critical to assessing the selectivity of the technique. Please provide the linear calibration data for DA in the presence of interferent (Figure 8a) and compare to response with DA alone.

Minor

• Please state the specific “computational routines for filtering and smoothing” used in this study.

• How many individual electrodes were tested for both in vitro and in vivo testing? How many animals were used for in vivo testing?

• How is “accuracy” determined when comparing calculated charges from FSCAV limits and curvature limits to the “True Charges”? Does this refer to linear sensitivity? Please be more specific.

• It is notable that the DA, NEP and EPI experiments in Figures 2, 3 and 4 reflect different experimental designs. Why was NEP and EPI tested up to 5 µM whereas DA was only tested to 1 µM? Why were there so few replicates for NEP and EPI as compared to DA? If the authors intend to provide a fair comparison between the responses of various chemicals then the experimental protocols should mimic each other.

• The authors state that “Figure 4 shows an example were curvature integration boundaries provide the best results among the four methods considered”. I only see 3 methods represented (please include the “True” results. Also, what does “best results” mean?

• Please show the peak shape for NEP since the peak shape is shown for DA and EPI.

• Why are average values for the inflection method not displayed on Figures 3 and 4?

• Please provide markings for all four of the limits on Figure 4a (similar to Figure 1a)

• The a) and b) labeling on Figures 1 and 5 are not consistent with the other figures.

• How do the in vivo limits compare to the previous limits defined by FSCAV? Please incorporate those cutoff values into Figures 5a and 5b.

• Average ± What? Is it standard deviation? Standard error?

• What do the numbers in the legend of Figure 6a stand for?

• The authors state that there cannot be a “True Limit” for the reduction peak. They previously stated that the switching potential could serve as a limit for the “True” method in cases where the peak does not return to zero (i.e. in vivo data). Wouldn’t this justification be applicable here as well and allow for inclusion of the “True” data set into Figure 7?

• It is confusing why the two background-subtracted voltammograms for EPI (Figure 4a and Figure 8b) look so different. Are they at different concentrations? Why is the ratio of the two oxidation peak heights so drastically different between the two responses?

Reviewer #2: This paper presents a novel approach to identify neurochemicals and a clever effort to reduce false observations in FSCV or FSCAV data. Given that most of voltammetric post-analysis methods are prone to misinterpretation, a more mathematical approach – like the one presented in this work – can contribute to making more accurate statements on the collected biological data. It would be interesting to see how this work would develop in future.

The paper is well written and the title fits the premises of the target study. I have several rather technical comments to further improve the clarity of the presented method/results (listed here in the order of appearance).

Abstract-------------------------------

1. In the last paragraph, the authors claimed, “Our results show that these methods allow quantification of oxidation and reduction reactions, for multiple analytes, both in vitro and in vivo.” however:

a) the provided results in the manuscript do not fully address in vivo quantification of the reduction reactions (pages 10,3.3). Please correct the text or include more data to support the claim.

b) the provided data in the manuscript do not support the claim on detection of “multiple analytes” in vivo (see page 10, section 3.2, Figure 6). Please amend the text to better reflect the result or include more data to support the text.

Method--------------------------------

2. Please specify what is meant by “A 110 µm CFM”. Both the length and diameter of the used electrode should be provided (page 2, 2.1)

3. The authors have presented this work as an automatic quantification approach however, the manuscript is missing details on how theses boundary conditions were implemented into an automatized system. Authors should support the concept with a schematic flowchart/algorithm of the applied “routines”. The flowchart should provide an overview of the applied computational procedure for filtering/smoothing and background- subtractions in addition to the correction routines (page 5, 2.3).

Results and discussion-------------------------------

4. According to the manuscript, 70 injections of DA and 15 injections of norepinephrine were applied to produce the results (page 5, 3.1). However, it is not clear how many electrodes were used to generate the presented data. Please provide the sample size, i.e. number of electrodes in addition to the number of injections per electrode, for the presented results.

5. Statistical analysis (including the level of significance) should be included to support the presented comparisons. An optical judgment is indeed a good cue but not sufficient for deciding on the efficacy of the presented method. Here are some example statements which should be statistically backed up:

“In Figure 4 shows an example were curvature integration boundaries provide the best results among the four methods considered” (page 8, 3.1).

“Figure 3(b) shows that oxidation charges obtained with curvature limits provide a better approximation to true charges than FSCAV limits.” (page 7, 3.1)

“…curvature boundaries select a wider region than that enclosed by the injection boundaries (figure 1), and they provide a better approximation to the charges obtained by using true boundaries,..” (page 7, 3.1)

6. Please include one paragraph discussing the limitations and challenging aspects of the proposed approach.

6. PLOS authors have the option to publish the peer review history of their article (what does this mean?). If published, this will include your full peer review and any attached files.

Reviewer #1: No

Reviewer #2: No

---

## [Author Response · Author response to Decision Letter 0]

28 Apr 2021

All comments have been addressed in the revised manuscript and "response to reviewers" document. Thank you for your consideration.

---

## [Decision Letter · Decision Letter 1]

29 Jun 2021

Integral methods for automatic quantification of fast-scan-cyclic-voltammetry detected neurotransmitters

PONE-D-20-25050R1

Dear Dr. Asp,

We’re pleased to inform you that your manuscript has been judged scientifically suitable for publication and will be formally accepted for publication once it meets all outstanding technical requirements. As a final improvement to your paper, please consider the comment from Reviewer 1 regarding the figure formatting (colours and font size)

Kind regards,

Maria Asplund, PhD

Academic Editor

PLOS ONE

Additional Editor Comments (optional):

Reviewers' comments:

Reviewer's Responses to Questions

**Comments to the Author**

1. If the authors have adequately addressed your comments raised in a previous round of review and you feel that this manuscript is now acceptable for publication, you may indicate that here to bypass the “Comments to the Author” section, enter your conflict of interest statement in the “Confidential to Editor” section, and submit your "Accept" recommendation.

Reviewer #1: (No Response)

Reviewer #2: All comments have been addressed

2. Is the manuscript technically sound, and do the data support the conclusions?

Reviewer #1: Yes

Reviewer #2: Yes

3. Has the statistical analysis been performed appropriately and rigorously? 

Reviewer #1: Yes

Reviewer #2: Yes

4. Have the authors made all data underlying the findings in their manuscript fully available?

Reviewer #1: Yes

Reviewer #2: Yes

5. Is the manuscript presented in an intelligible fashion and written in standard English?

Reviewer #1: Yes

Reviewer #2: Yes

6. Review Comments to the Author

Reviewer #1: The authors have adequately addressed my concerns, and as a result the manuscript is much improved.

My only remaining comment is regarding the formatting of figures. The text on the figures is difficult to read at times due to either the small font, grey text coloring or a combination of both. Please adjust the figure text so that it is easier to read.

Reviewer #2: (No Response)

7. PLOS authors have the option to publish the peer review history of their article (what does this mean?). If published, this will include your full peer review and any attached files.

Reviewer #1: No

Reviewer #2: No

---

## [Editor Report · Acceptance letter]

14 Jul 2021

PONE-D-20-25050R1 

Integral methods for automatic quantification of fast-scan-cyclic-voltammetry detected neurotransmitters 

Dear Dr. Asp:

I'm pleased to inform you that your manuscript has been deemed suitable for publication in PLOS ONE. Congratulations! Your manuscript is now with our production department. 

Kind regards, 

on behalf of

Dr. Maria Asplund 

Academic Editor

PLOS ONE